# The Abrasive Effect of Moon and Mars Regolith Simulants on Stainless Steel Rotating Shaft and Polytetrafluoroethylene Sealing Material Pairs

**DOI:** 10.3390/ma17174240

**Published:** 2024-08-27

**Authors:** Gábor Kalácska, György Barkó, Hailemariam Shegawu, Ádám Kalácska, László Zsidai, Róbert Keresztes, Zoltán Károly

**Affiliations:** 1Institute of Technology, Szent István Campus, Magyar Agrár- és Élettudományi Egyetem (MATE), Páter K. u. 1., H-2100 Gödöllő, Hungary; kalacska.gabor@uni-mate.hu (G.K.); barko.gyorgy.csaba@uni-mate.hu (G.B.); zsidai.laszlo@uni-mate.hu (L.Z.); keresztes.robert.zsolt@uni-mate.hu (R.K.); 2Mechanical Engineering Doctoral School, Magyar Agrár- és Élettudományi Egyetem (MATE), Páter K. u. 1., H-2100 Gödöllő, Hungary; shegawh@yahoo.com; 3Laboratory Soete, Tech Lane Gent Science Park—Campus A, University of Gent, Technologiepark 131, B-9052 Gent, Belgium; 4HUN-REN Research Centre for Natural Sciences, Institute of Materials and Environmental Chemistry, Magyar Tudósok Krt. 2, H-1117 Budapest, Hungary

**Keywords:** lunar regolith, Martian regolith, abrasive test, wear, shaft and seal

## Abstract

For space missions to either the Moon or Mars, protecting mechanical moving parts from the abrasive effects of prevailing surface dust is crucial. This paper compares the abrasive effects of two lunar and two Martian simulant regoliths using special pin-on-disc tests on a stainless steel/polytetrafluoroethylene (PTFE) sealing material pair. Due to the regolith particles entering the contact zone, a three-body abrasion mechanism took place. We found that friction coefficients stabilised between 0.2 and 0.4 for all simulants. Wear curves, surface roughness measurements, and microscopic images all suggest a significantly lower abrasion effect of the Martian regoliths than that of the lunar ones. It applies not only to steel surfaces but also to the PTFE pins. The dominant abrasive micro-mechanism of the disc surface is micro-ploughing in the case of all tests, while the transformation of the counterface is mixed. The surface of pin material is plastically transformed through micro-ploughing, while the material is removed through micro-cutting due to the slide over hard soil particles.

## 1. Introduction

The exploration and exploitation of the resources of the nearest planetary bodies are crucial to advancing humanity’s presence in space, offering significant benefits for scientific research, space travel, and extra-terrestrial settlements. Among the numerous challenges posed by the planetary environment, the abrasive properties of extra-terrestrial dust represent a critical area of investigation that has yet to be fully explored. Extra-terrestrial dust, consisting of fine regolith particles with complex morphologies, poses a significant risk to machinery and human health mainly due to its abrasive and adhesive properties [1,2]. The behaviour of regolith can impact transportation on Mars and Moon. Regolith on Mars is generally fine-grained and powdery, containing a mix of dust and sand. Due to Mars’s lower gravity than Earth, the regolith particles are less compact and can behave more like loose sand. These particles are transported by wind and can accumulate in plains, valleys, and impact craters. Dust storms on Mars can cover large regions and last for days or even months. Unlike Earth, Mars does not have widespread vegetation or plant life that helps anchor the soil and prevent erosion [3]. The absence of vegetation means fewer mechanisms to stabilise the soil and reduce dust generation. The dusty nature of Mars presents unique challenges for future human missions and robotic exploration, especially for wheeled vehicles to gain traction, and can cause wheels to sink or slip [4,5].

Lunar regolith comprises particles spanning from sub-micron dust to larger grains, contributing to the diverse texture of the regolith located on the lunar surface. The absence of a lunar atmosphere allows this dust to persist without abrasion and to retain sharp edges that can damage materials and infiltrate systems, causing premature wear and failure of mechanical components [6]. Given the potential for dust exposure, materials need special properties to function in the dust and regolith of Mars and the Moon to provide effective sealing and insulation to protect internal components and maintain environmental control.

The abrasion type of wear on the materials caused by regolith simulants has been studied by abrasion tests carried out in traditional tribological or custom-made setups with lunar and Martian simulants as a third body. Ishibashi et al. [7] concluded that the hardness and fracture toughness of materials fundamentally determine their wear characteristics. Materials with K_IC_ less than 4 MPa.m^−1/2^ give rise to a high wear rate, while materials with K_IC_ greater than 6 MPa.m^−1/2^ considerably reduced the wear rates. Matsumoto et al. [8] pointed out that the coefficient of friction (CoF) was significantly lower for aluminium (Al 6061) than steel (SS 440C) specimens. However, its value was hardly affected by the presence and size of the simulants. The wear mechanism for the harder steel involves adhesion and particle transfer. For the softer aluminium, wear can be attributed to abrasive wear caused by the third body and adhesive wear. Larger particles have a tendency to plough into the specimens in contact and cause material removal during sliding. Besides the traditionally used steel and aluminium grades overviewed and detailed formerly [9], options to mitigate or avoid wear problems include the application of diamond-like coatings with extra-high hardness and, therefore, superior wear resistance or MoS_2_ coatings with low coefficient of friction on metallic parts [10,11], as well as the protection of friction surfaces from dust by various low-friction sealants [12]. The tribological performance of aerospace materials in the lunar environment has been summarised in a recent review [13]. Delgado et al. reported the outstanding performance of PTFE sealants when metallic materials were exposed to the abrasive effects of lunar simulant powders [14]. However, the incoherence between the wear of insulating materials and surface roughness still needs to be answered. The importance of developing durable technologies that can withstand the harsh planetary environment cannot be overstated, as it is crucial for the overall success of long-duration space missions [15].

Recently, the authors have carried out extensive, systematic studies to investigate the tribological properties between different metals selected together by experts of the European Space Agency (ESA) that have high potential being used in space missions and lunar and Martian dusts, using custom-developed/modified equipment as well as the effectiveness of various insulating materials. This paper reports the results of tribological properties of lunar and Martian simulant powders acting on stainless steel shaft material. Through a comprehensive review of existing studies and identifying research gaps, we strongly advocate for a multidisciplinary approach that combines materials science and mechanical engineering to develop solutions that mitigate the adverse effects of lunar and Martian dust on space exploration efforts.

For ground conditions and materials, the characteristics, influencing factors, and damage mechanisms of abrasive wear are known to an acceptable level in tribological practice. Based on this knowledge, the abrasive effects of dust and soil in space applications should also be explored for machinery parts at risk. The chemical composition, particle size, and shape distribution of regolith are highly variable [16]. Their differential wear effects on rotating shafts and seals need to be better understood. Here, we studied individual wear tribological systems with selected regolith and shaft/seal material pairs under ground laboratory conditions. Based on the frictional wear and microscopic results, the damage of each tribological system can be ranked in relative order and provide input information for further, much more costly, and complex wear investigations (component testing, gas and vacuum environments, extreme temperature effects). The adhesion and grain adhesion of each regolith in the test system was also analysed and revealed, which has a significant influence on the tribological systems.

## 2. Materials and Methods

### 2.1. Selection of Shaft and Sealing Materials

Material selection was based on a recent study [9] on the individual mechanisms where an abrasion problem can occur in space applications, analysing the structural material and sealing combinations tested and used so far and the frequency of occurrence of each solution. Accordingly, the shaft material selected for the current research and approved by ESA was 316 L (ASTM) or 1.4404 (EN) steel grade due to mainly their austenitic structure, thus, low-temperature applicability, strength, and machinability. For sealing material, a natural PTFE single lip seal (Ecoflon 1, spring-loaded) with a Ro1-AS profile has been selected for testing according to ESA preferences.

### 2.2. Mars and Moon Soil Simulants Selected for Abrasion Tests

An extensive review of materials and simulants available and detailed criteria of selection for testing have been described in [9]. The selection criteria of simulants are considered as follows:Mineralogical form and particle shape of the simulant are more critical than chemical composition;Simulant should represent the landing site and resemble the actual samples as closely as possible;Level of mineralogical fidelity: particle shape relevant to the abrasion test, allowing good representativity of the test with real lunar/Martian regolith;Particle size distribution.

Considering the abovementioned objectives, two regolith simulants have been selected to characterise Mars. General Martian soil has been represented by Exolith MGS-1 (Exolith Lab, Orlando, FL, USA), and a particular Martian location, the soil of Jezero crater, was characterised by Exolith JEZ-1. Two simulants have been chosen for the Moon. Lunar Mare soil has been characterised by Exolith LMS-1, and lunar Highland soil has been demonstrated by Exolith LHS-1. Significant characteristics of the selected regoliths are listed in Table 1, while the detailed characterisation of the simulants is available in [17,18,19,20].

### 2.3. Testing and Measurement Methods

Measurements were conducted at the coupon level. The tribo-tester employed was a suitably modified pin-on-disc device, with the disc’s surface uniformly covered with regolith.

The method depicted in Figure 1 represents a controlled test in accordance with DIN 50322 category VI. This system facilitates the examination of the intricate mechanisms involved when simulated materials with flat surfaces interact through sliding. It allows for the comprehensive investigation of both frictional and three-body abrasion mechanisms, as well as the assessment of contact zone deformation and abrasion. Additionally, it enables the observation of micro-geometric alterations in the surfaces under investigation. In order to analyse the embedding capacity of the simulants, their abrasive impact, their adhesion within the contact zone, and the influence of grain dynamics, scanning electron microscopy (SEM, Zeiss EVO 40, Zeiss, Jenna, Germany) and energy-dispersive X-ray spectroscopy (EDX, JEOL JSM-IT700HR, JEOL, Tokyo, Japan) measurements are employed. A detailed exposition of the system’s specific settings is valid.

Pin material: natural PTFE (Ln) as 8 × 8 × 8 mm cube;Rotating disc (*n* = 0.6 1/s) material: stainless steel 1.4404 (Ss)Sliding speed on the pin’s centre line: v = 0.1 m/s, on 25 mm sliding path radiusStress from normal load: 0.2 MPa;Ambient temperature: 22–24 °C;Relative humidity RH: 40–50%;Start: in clean contact;Sliding path: covered with abrasive simulants;Type of regolith applied: LHS-1, LMS-1, MGS-1, JEZ-1.

The results presented of the pin-on-disc system stem from an experimental setup. According to the concept, a cut piece of sealing material 10 × 10 × 10 mm—as a pin—slides in an annular path with constant speed and load where the material of the rotating disc is the same as that of the modelled shaft. A boundary ring is attached to the disc’s edge because the disc’s surface is freely covered with abrasive particles of 3 mm thickness. During the rotation of the disc, the diverting plates continuously push the regolith particles back to the sliding track. During sliding, the particles enter between the shaft material (disc)/seal surface pairs and exert an abrasive effect, as can occur with a real rotating shaft/seal model contact. During the measurement, the change in the abrasive friction resistance (similar to the change in the torque in the case of a rotating shaft) and the wear with layer transformation of the seal and disc material surface are measured on-line.

The discs/pin frictional pairs were tested under four different types of regolith. In each layout, four repeated runs of 2, 6, 15, and 30 min were performed with fresh pins on a fresh metal disc surface.

Friction force and the vertical displacement of the pin’s holder head (referred to as wear hereinafter) were measured on-line during the pin-on-disc tests using a Spider 8 data acquisition system. The 2D and 3D topography of the disc and the pin were also measured using a Keyence VR 5200 3D optical microscope (white light interferometer).

## 3. Results

### 3.1. On-Line Friction and Wear

Figure 2 shows the frictional behaviour of the various simulant powders on a steel disc. The friction coefficient was a calculated abrasive friction number characteristic of an open three-body abrasive sliding mechanism. The graphs are plotted with a moving average of 60 data. The abrasive tests were carried out for various durations (2/6/15/30 min) in each examined material pair, and they typically show excellent reproducibility (Figure A1, Figure A2, Figure A3, Figure A4, Figure A5, Figure A6, Figure A7 and Figure A8, Appendix A). However, for the sake of visibility, only the plots for the longest duration are shown here for each simulant powder.

At the beginning of the friction curves, the effect of the particles entering the contact zone and the formation of the three-body abrasion mechanism can be observed. In the first 8–20 m of the friction process, the run-in phase of the PTFE/steel adhesive connection reaches a maximum friction value varying between 0.30 and 0.40; however, the trend of the run-in processes differs at each system.

In the case of lunar simulants, after 10–20 m, the regolith particles already enter the contact zone, and the adhesion resistance changes to a lower sliding resistance due to the grain rolling. Then, in the operation of the Berthier-Eleőd three-body abrasion [21,22,23], the dynamic zone equilibrium can be seen on the corresponding curves. The fluctuation of the friction peaks returns according to the contact zone mechanism. The particles get in, are partially embedded in the softer PTFE surface, and adhere to each other in several layers, as well as roll on the surface, break up, and sometimes become congested with the formation of larger particles and blocks (returning peaks in the curves). Meanwhile, the effect of sliding–rolling grain marks also appears on the surface of the steel: abrasive scratches and the phenomenon of grain embedment. In the case of LHS-1, the friction stabilises around 0.2, showing an almost constant or slightly decreasing sliding resistance. This coefficient of friction is less than half of that obtained for titanium/PEEK (0.52) or titanium/ATSP (0.41) tribopairs with LHS-1 simulants in a recent study [24]. In contrast, the LMS-1 simulant shows no stabilisation, and the abrasive sliding resistance varies between 0.2 and 0.3.

For the MGS-1 Martian simulant dust, the particles in the formed layer start to shear and roll, causing a temporary decrease in the resistance after 35–45 m, but the friction resistance is different from that of the lunar simulant powders. The decrease in friction that occurs after the running-in phase does not lead to stabilisation but shows a slightly increasing trend. In accordance with the results of the wear curve (Figure 3) and the surface photos, the layer adhering to the metal surface is more uneven, the migration and jamming of the particles increase the frictional resistance during the stable stage of the process, and it stabilises at a value of around 0.35–0.4. The effect of sliding–rolling grain marks also appears on the surface of the steel: abrasive scratches and the phenomenon of grain embedment (Figure 4c and Figure 5c). The JEZ-1 simulant behaved fundamentally differently compared to the other three powders. Transient processes can also be observed within the run-in phase, which can be explained by the entry of dust, its temporary adhesion, and the effect of modifying the initial adhesion. After the run-in phase, relatively few particles can stick in the contact zone, which is also clearly visible in the surface photographs (Figure 4). In the stable phase of the sliding, the frictional resistance shows a linear decreasing trend, and the abrasive friction coefficient decreases to 0.2.

Figure 3 illustrates the abrasive wear of the shaft material with respect to the travelled distance. The wear is measured by the pin holder head’s vertical displacement, including the samples’ real wear, the deformation, the heat expansion, the migration, and the embedding of the abrasive particles in the contact zone. After the period of adhesive sliding, the regolith entering the contact zone raises the pin holder head, resulting in negative wear after around 10, 20, and 30 m for LHS-1, LMS-1, and MGS-1 simulants, respectively. The extreme peak values (down to −0.7 mm) indicate the effect of the accumulation of larger particles. As a result of the dynamic three-body zone mentioned in the case of friction, the movement of grains and flakes can be perceived by the local peaks of the curves. For all four systems, the dynamic balance that develops after the run-in phase causes a linear change in the stable section of the wear curve. In the case of LHS-1 and LMS-1, the linear section, i.e., the abrasive wear rate, increases with almost the same slope. The difference is that in the case of LMS-1, a much thicker regolith layer accumulates in the contact zone at the beginning of the wear process, so the wear curve reaches the region of positive wear later than in the case of LHS-1. The MGS-1 and JEZ-1 regoliths showed much smaller particle embedment at the end of the run-in, and their linear wear curves are also different. MGS-1 resulted in a low-slope, nearly horizontal wear curve, while in the case of JEZ-1, there was no measurable significant wear change in the system.

### 3.2. Surface Analyses

The microscopic photographs (Figure 4) confirmed that in the case of LHS-1 and LMS-1, significant dust layer adhesion and embedment were observed on steel surfaces. SEM images of the steel surface (Figure 5) after abrasion tests also suggest a more severe abrasion of the lunar regoliths than Martian ones.

EDX analyses found on the Ss surface in the case of LHS-1 that larger embedded particles are mainly composed of calcium–aluminium–silicates, while plenty of smaller particles contain sodium and magnesium as well (Figure 6). With LMS-1 regolith on the Ss surface, plenty of tiny grooves or pits are discernible and are covered with nanosized or submicron-sized dust particles. Minor heterogenous distribution can be observed for Si, Mg, and Ti, indicating larger particles of Olivine and Ilmenite. On the other hand, MGS-1 and JEZ-1 formed a less adherent layer, and the regolith movement in the contact zone is also much smaller, according to the photographs. By EDX, particles of Olivine, Anorthosite, Basalt, or Pyroxene are embedded into the steel surface. Although the abrasive dust particles are composed mainly of Ca, Mg, Si, and Al, their presence is relatively small. On the PTFE specimen of JEZ-1 regolith, even surface machining marks can be discerned due to minimal final wear. On the wear curve, this can be identified with a value close to 0 “wear”. The reduced effect of the three-body mechanism is also characterised by the fact that, in the case of the JEZ-1 simulant, the fluctuation of the wear curves and the deviation of the measurement results were minimised.

The major surface roughness parameters, including S_a_ (arithmetic mean height), S_z_ (maximum height), S_q_ (root mean squared height), Skewness, and Kurtosis, are provided in Table 2. All measured parameters (S_a_, S_z_, S_q_) related to the surface height suggest that the steel disc’s roughness plummeted due to the abrasive effect of the simulant powders, irrespective of the type of simulant. The near-zero skewness (S_sk_) after the tests also suggests the disappearance of the dominance of the peaks. The steel surface was polished in the wear track, and the steel topography was transformed by abrasion. The decrease in the change in the total S roughness parameters confirmed this. The reduction in S values—i.e., microgeometry change, transformation—is smaller in the case of lunar simulants (except LHS-1 in S values) than in the case of Martian simulants that may suggest a milder abrasive effect of the former simulant. However, this is not the case. The reduced S values for lunar simulants can be attributed to the accumulated regolith layer by the end of the abrasive test, which is confirmed by the wear curves. In parallel, MGS-1 and JEZ-1 regoliths resulted in the greatest smoothing, i.e., the microgeometry transforming effect. The milder abrasive effect of the Martian regolith is probably due to the significantly lower proportion of glass-rich, high-hardness basalt among the phases that make up the regolith.

Surface topography data extracted from the 3D map were utilised to compute the degree of penetration (D_p_). Hokkirigawa et al. [25] introduced a formula predicting the penetration depth in a single-asperity contact scratch test, considering a specific normal load and indenter tip radius on a surface of known hardness. A critical D_p_ value of approximately 0.2 is established to delineate between micro-ploughing and micro-cutting across various attack angles.

Different ranges of D_p_ values correspond to each wear micro-mechanism, with the material hardness exerting influence on these ranges.

From the seal/shaft application, it is clear that low-stress or scratching abrasion occurs when lightly loaded abrasive particles impinge on and move across the wearing surface, producing cutting and ploughing on a microscopic scale. Abrasive wear has three different wear modes: micro-cutting (a), wedge forming (b), and ploughing (c), all shown in Figure 7. Wear particles are formed differently depending on these three modes. In all these three abrasive wear modes, grooves are formed due to wear particle generation and material plastic flow to form ridges on both sides of a groove [26].

The transition from micro-ploughing to micro-cutting occurs as the attack angle of the abrading particle increases, leading to a higher D_p_ [27,28]. D_p_ also serves as a partial indicator of wear severity, with the micro-mechanism itself aiding in distinguishing between mild and severe wear regimes.

In single-asperity testing, the topographic data and wear micro-mechanisms encountered by individual scratches within each groove are analysed. Initially, the 3D topographic data are extracted. To determine the D_p_ of the wear scar, cross-section profiles of the scratch are obtained at several defined locations across the groove. These extracted data are then averaged and consolidated into a single average cross-section profile (Figure 8) [29]. The groove depth is divided by half the width of the profile at the surface level to derive D_p_.

Following a topographic surface characterisation measurement, it becomes feasible to compute the mean D_p_ of all the grooves present on the worn surface. Coronado et al. [30] confirmed the validity of the following formula for D_p_ when comparing two abraded surfaces:(1)$$D_{p}=\frac{R_{z}}{\frac{R_{sm}}{2}}$$

R_z_ represents the ten-point height [µm], while R_sm_ [µm] denotes the mean spacing at the mean line. These values correspond to the groove depth and the groove width of a single scratch, respectively. Different D_p_ value ranges are associated with each wear micro-mechanism, depending also on the hardness of the material. Micro-ploughing, characterised by plastic deformation of the surface and significant ridge formation along the grooves, is considered dominant for low (<0.17) D_p_ levels. A transition (wedge formation) is followed with increasing D_p_ to eventually result in a pure micro-cutting mechanism (above ~0.35) with narrow and deep grooves and significant chip formation.

The abrasion grooves were analysed to determine D_p_ values. Based on multi-line groove analyses, the calculated D_p_ was always less than 0.1 (Table 3), which means that the dominant surface transformation of the steel surface is micro-ploughing in the case of all tests. In contrast, the larger than 0.1 D_p_ values for the PTFE pins indicate a transformation/mixed mechanism of abrasive wear. The surface is plastically transformed through micro-ploughing (distorted surfaces, ridges), while the material is removed through micro-cutting. The photos and SEM images of PTFE surfaces (Figure 9) clearly show that in the case of JEZ-1 (Martian) regolith, a much smaller amount of particles can stick in the contact zone. As a result, the final wear is minimal on the PTFE specimen, and even surface machining marks can be discerned as compared to LHS-1 (lunar) simulant dust.

## 4. Conclusions

The abrasive effect of two lunar (LHS-1, LMS-1) and two Martian (MGS-1, JEZ-1) simulants was investigated by pin-on-disc type test using stainless steel discs (1.4404) and polytetrafluoroethylene as pin material. Friction coefficients varied between 0.2 and 0.4 for all simulants. After the running-in phase, the friction stabilised with a slight decrease in the case of LHS-1 and JEZ-1, while no stabilisation was reached in the case of LMS-1 and MGS-1. The different regoliths of the same origin had rather similar abrasive behaviour, but there was a definite difference between the abrasive behaviour of the lunar and Martian simulants. A more significant dust layer adhesion and embedment were observed for lunar regoliths than for Martian ones. The wear effect of the lunar dusts was considerably higher than Martian ones according to wear curves and surface roughness measurements, confirmed by SEM, too. Martian regoliths showed much smaller particle embedment, too, at the end of the run-in phase. MGS-1 resulted in a low-slope, nearly horizontal wear curve, while in the case of JEZ-1, there was no measurable significant wear change in the system. The steel surface was polished in the wear track, and the steel topography was transformed by abrasion, confirmed by the decrease in the surface roughness parameters. The dominant surface transformation of the steel surface was micro-ploughing in all tests. The results show that lunar conditions present a higher challenge for protecting rotating parts from the abrasion effects of the prevailing planetary dust. To avoid more intense abrasive effects caused by Moon regoliths, the use of alternative metallic and non-metallic materials for moving parts should be investigated.

## Figures and Tables

**Figure 1 materials-17-04240-f001:**
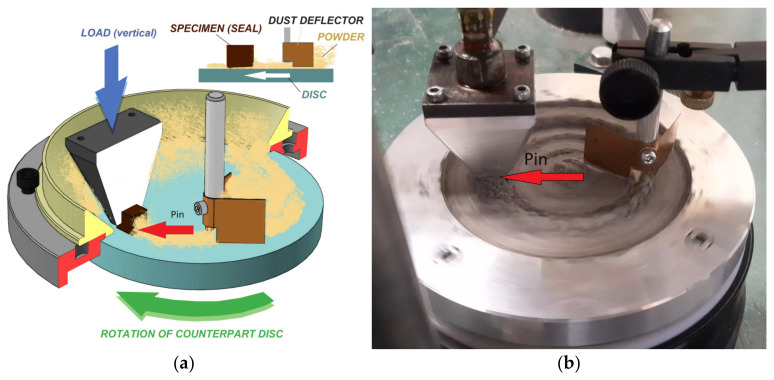
Theory and practice of modified pin-on-disc measurements: (**a**) schematic of the layout; (**b**) the retainer blades cover back the path with regolith.

**Figure 2 materials-17-04240-f002:**
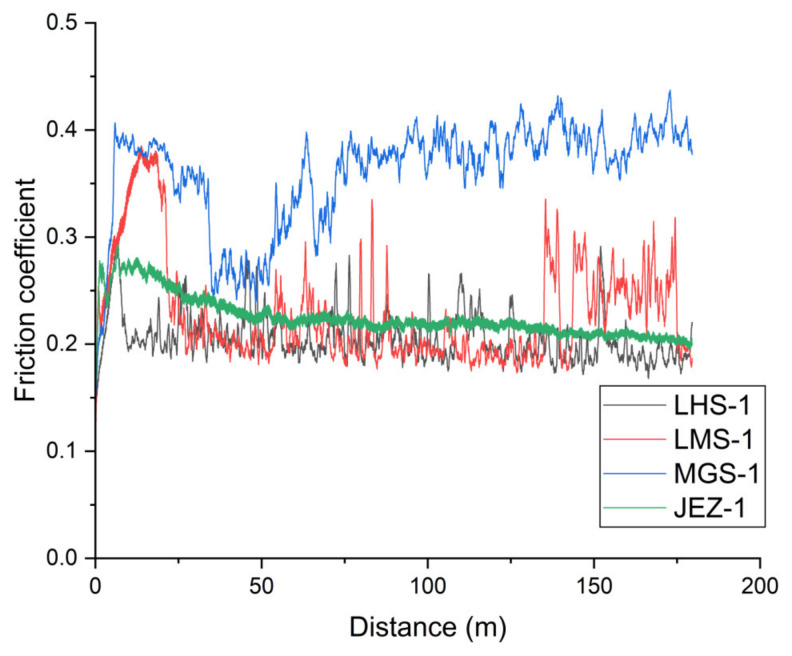
Coefficient of friction of frictional surfaces as a function of travelled distance for lunar (LHS-1, LMS-1) and Martian regoliths (MGS-1, JEZ-1).

**Figure 3 materials-17-04240-f003:**
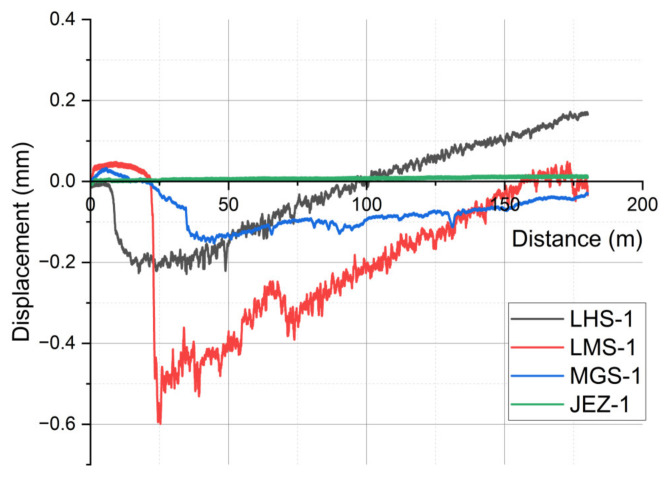
Wear behaviour of frictional surfaces as a function of travelled distance for lunar (LHS-1, LMS-1) and Martian regoliths (MGS-1, JEZ-1).

**Figure 4 materials-17-04240-f004:**
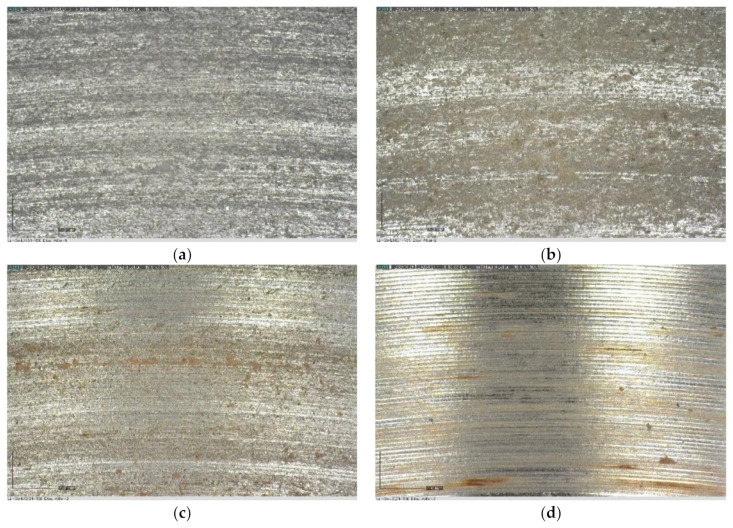
Light microscopy images of steel discs after 30 min of abrasion test with (**a**) LHS-1, (**b**) LMS-1, (**c**) MGS-1, and (**d**) JEZ-1 regolith simulants: lunars are significantly covered and embedded while Martians are less damaged.

**Figure 5 materials-17-04240-f005:**
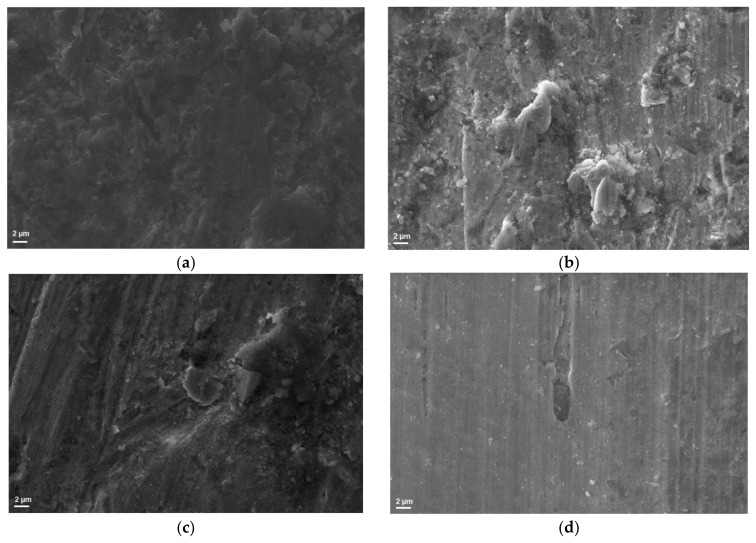
SEM images of steel discs after 30 min of abrasion test with (**a**) LHS-1, (**b**) LMS-1, (**c**) MGS-1, and (**d**) JEZ-1 regolith simulants.

**Figure 6 materials-17-04240-f006:**
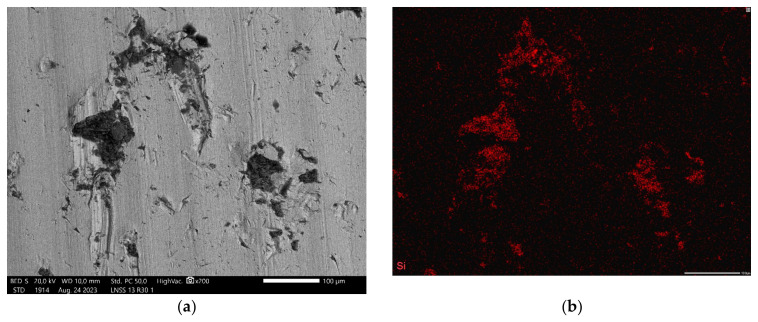
A selected (**a**) SEM image and EDX elemental mapping for (**b**) Si, (**c**) Al, (**d**) Ca, (**e**) Mg, and (**f**) Na elements on the steel disc after 30 min of abrasion test with LHS-1 regolith simulant as abrasive media.

**Figure 7 materials-17-04240-f007:**
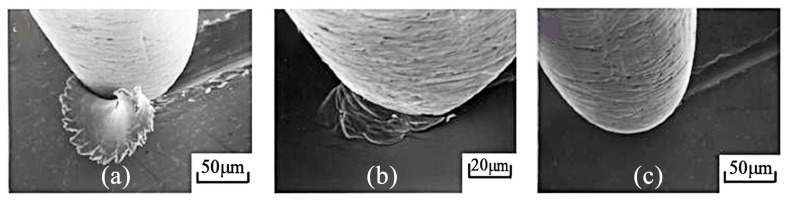
Different modes of abrasive wear on SEM images: cutting mode (**a**) steel pin on brass plate, wedge-forming mode (**b**) steel pin on stainless steel plate, ploughing mode (**c**) steel pin on brass plate [22,23].

**Figure 8 materials-17-04240-f008:**
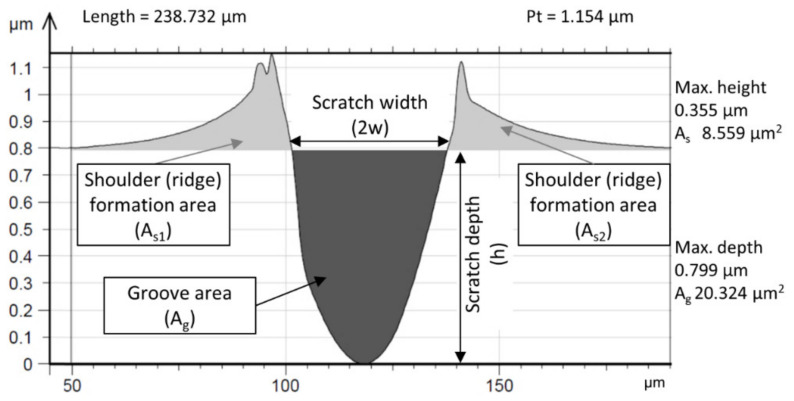
Extraction of groove data from an abrasion scratch test [29].

**Figure 9 materials-17-04240-f009:**
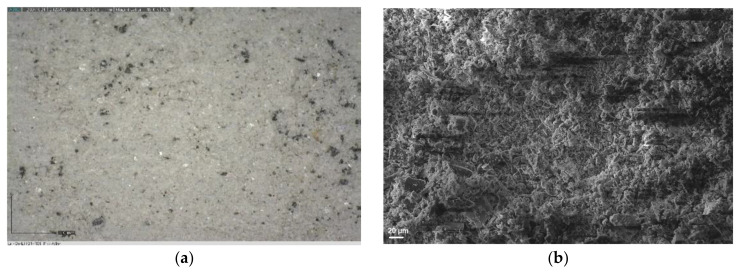
Photos (**a**,**c**) and SEM images (**b**,**d**) of PTFE surfaces after 30 min of abrasion test with LHS-1 (upper row) and JEZ-1 (lower row) regolith simulants.

**Table 1 materials-17-04240-t001:** Significant characteristics of the regolith simulants.

Regolith Simulants	LMS-1	LHS-1	MGS-1	JEZ-1
Original Material	Average Lunar Mare Soil	Average Lunar Highland Soil	Average Martian Soil	Martian Soil of Jezero Crater
Phase Composition	Pyroxene, 32.8% Glass-rich basalt, 32.0% Anorthosite, 19.8% Olivine, 11.1% Ilmenite, 4.3%	Anorthosite, 74.4% Glass-rich basalt, 24.7%	Anorthosite 27.1% Glass-rich basalt 22.9% Pyroxene 20.3% Olivine 13.7% Mg-sulfate 4.0% Ferrihydrite 3.5% Hydrated silica 3.0% Magnetite 1.9% Anhydrite 1.7% Fe-carbonate 1.4%	Olivine 32.0% Anorthosite 16.0% Glass-rich basalt 13.5% Pyroxene 12.0% Mg-carbonate 11.0% Smectite 6.0% Mg-sulfate 2.4% Ferrihydrite 2.1% Hydrated silica 1.8% Magnetite 1.1% Anhydrite 1.0%
Physical Properties	Bulk density: 1.56 g/cm^3^ Median particle size: 60 μm Particle size range: >0.04–1000 μm Particle shape: sub-rounded to angular	Bulk density: 1.30 g/cm^3^ Median particle size: 60 μm Particle size range: >0.04–1000 μm Particle shape: sub-angular to angular	Bulk density: 1.29 g/cm^3^ Median particle size: 60 μm Particle size range: >0.04–600 μm	Bulk density: 1.54 g/cm^3^ Median particle size: 60 μm Particle size range: <0.04–500 μm
Reference	[18]	[17]	[19]	[20]

**Table 2 materials-17-04240-t002:** Surface roughness parameters of the steel disc after 30 min test period.

3D Parameters, µm		Regolith Simulants			
		LHS-1	LMS-1	MGS-1	JEZ-1
S_a_	Pristine	1.01	1.02	1.02	1.02
	Worn	0.79	0.89	0.70	0.71
	Difference, %	−22	−12	−32	−30
S_z_	Pristine	168.24	171.64	171.64	166.87
	Worn	18.69	19.22	14.13	9.32
	Difference, %	−89	−89	−92	−94
S_q_	Pristine	1.72	1.85	1.85	1.917
	Worn	1.01	1.16	0.89	0.88
	Difference, %	−41	−38	−52	−54
Ssk	Pristine	2.34	3.65	3.65	3.82
	Worn	−0.19	0.05	−0.32	0.09
	Difference, %	−108	−99	−109	−98
Sku	Pristine	159.73	179.34	179.34	185.56
	Worn	4.31	4.61	4.31	2.80
	Difference, %	−97	−97	−98	−98

**Table 3 materials-17-04240-t003:** Degree of penetration values for the steel disc and the pins after 30 min test period.

D_p_, (R_z_/0.5 R_sm_)	Regolith Simulants			
	LHS-1	LMS-1	MGS-1	JEZ-1
Steel Disc	0.043	0.024	0.041	0.048
PTFE Pin	0.197	0.239	0.259	0.158

## Data Availability

The raw data supporting the conclusions of this article will be made available by the authors upon request.

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
