# Peer review of "The Abrasive Effect of Moon and Mars Regolith Simulants on Stainless Steel Rotating Shaft and Polytetrafluoroethylene Sealing Material Pairs"

_materials, 2024, doi:10.3390/ma17174240_

Round 1
Reviewer 1 Report
Comments and Suggestions for Authors
It is a particularly interesting scientific work that can be a starting point in the dimensioning of future space equipment. The work is scientifically well founded. The experimental part is also solid, based on correct assumptions.
The conclusions are logical and relevant
Author Response
Comment: It is a particularly interesting scientific work that can be a starting point in the dimensioning of future space equipment. The work is scientifically well founded. The experimental part is also solid, based on correct assumptions.
The conclusions are logical and relevant
Answer: Thank you for the positive comment.
Reviewer 2 Report
Comments and Suggestions for Authors
This paper is majorly about the tribological behaviour of stainless steel and PTFE using pin-on-disc with abrasive particles simulating the regolith dust on Moon and Mars, it is an interesting topic and the authors obtained a lot of data and analyse the tribological mechanisms. However, this paper needs a major modification to clarify. Detail suggestions include:
1. The introduction is too long and need to be simplified.
2. What is ESA ESTEC?
3. The compositions of the tribo-pair need to be given in 2.1.
4. In 2.2, although the principle of selections of the soil simulants are listed, the four materials used in this research should be tabulated and clarified: composition, size distribution, shape of particles etc.
5. In 2.3 the dimensions of the pin should be given. Rotation speed of the disc and the diameter of the track to calculate the sliding speed should be presented as well. In figure 1, which one is the pin?
6. Figure 3 and the relevant description should be in discussion part about the tribological mechanisms analysis.
7. The detail of the surface roughness (appeared in table 1, Sa,Sz and Sq) and their relevant meaning, the standard used should be given in this part.
8. In figure 6, please explain the meaning and indication of negative and positive values? Negative =track? Positive =ridge or particle effect?
9. Figure 8, please point out which is embedded particle and the relevant EDX analysis, this is critical to explain what happen in the friction coefficient change? You observed the change of CoF at different time, have you observed the track difference?
10. Similarly, in figure 9, the EDX analysis of the particles on the pin to prove the transfer of materials is essential. By the way, what is C and D in Figure 9.
11. Detail information about ref 18-21 should be given.
Comments on the Quality of English LanguageGenerally, it is ok and understanble.
Reviewer 3 Report
Comments and Suggestions for Authors
Comments and Suggestions for Authors.
The document is interesting and shows different results. However, some adjustments must be considered to improve the quality of the document.
1. The paper presents in the introduction section an extensive review of the environmental conditions on the surface of the moon and mars. However, the authors do not include in the introduction section studies of other authors related to abrasion, wear and friction of materials caused by environmental conditions such as dust on the surface of the moon and mars available in literature.
2. Materials and Methods section the authors express the following: “We made and published a detailed study [4] of the individual mechanisms where an abrasion problem can occur in space applications. We analyzed the structural material and sealing combinations tested and used so far, and the frequency of occurrence of each solution”. This statement must be moved to the introduction section.
3. “Extensive review of materials and simulants available, and detailed criteria of selection for testing has been described in .” In lines 204-205, the reference is missing.
4. “Tests, devoted to investigate selected shafts and sealing materials, requiring abrasive media.” The statement makes no sense, or it is incomplete. Please check the entire document carefully or use a professional language editing service.
5. For the test pin on disc conditions the authors mention the following “Normal load: 0.2 MPa”. However, normal load in pin on disc test must be given in N due to MPa refers to stress generated by the contact pin area and disc.
6. “Figure 1. Theory and practice of modified pin-on-disc measurements his is a figure.” The caption of Figure 1 makes no sense. Please check the entire document carefully or use a professional language editing service.
7. Figure 2 is not necessary. The author can describe that four runs of 2, 6, 15, and 30 min were performed in the pin on disc test.
8. “Abrasive wear has three different wear modes, micro cutting (a), wedge forming (b), and ploughing (c), all shown in Figure 3. Wear particles are formed differently depending of these three modes. In all these three abrasive wear modes, grooves are formed as the result of wear particle generation and plastic flow of material to form ridges on both sides of a groove [23].” This statement must be moved to the results and discussion section.
9. “EDX analyses found on Ss surface in case of LHS1 that larger embedded particles are mainly composed of calcium-aluminium-silicates, while plenty of smaller particles contain sodium and magnesium as well”. The authors mention composition of EDX, however the Figure with the analysis is missing.
Comments on the Quality of English LanguageSeveral statements do not make sense or are incomplete. Review the entire document carefully or use a professional proofreading service.
Round 2
Reviewer 2 Report
Comments and Suggestions for Authors
In Table 1, please use the same code for LHS-1, LMS-1, MGS-1, and JEZ-1 as used in other places. Please carefully check the corresponding reference number is correct, for example, LMS-1 is Ref 17, not Ref 19!
Detailed information about Ref18-21 should be given.
Comments on the Quality of English LanguageNo
Author Response
Comments 1: In Table 1, please use the same code for LHS-1, LMS-1, MGS-1, and JEZ-1 as used in other places. Please carefully check the corresponding reference number is correct, for example, LMS-1 is Ref 17, not Ref 19!
Response 1: We have corrected and employed identical code for the regoliths thrughout the manuscript. We have also checked the reference numbers.
Comments 2: Detailed information about Ref18-21 should be given.
Response 2: We have added more information about Refs 17-20 in the References.
Reviewer 3 Report
Comments and Suggestions for Authors
The authors have corrected the comments satisfactorily to increase the quality of the document.
Author Response
Comments 1: The authors have corrected the comments satisfactorily to increase the quality of the document.
Response 1: Thank you for your previous comments and remarks.